# The Interplay between Glioblastoma and Its Microenvironment

**DOI:** 10.3390/cells10092257

**Published:** 2021-08-31

**Authors:** Mark Dapash, David Hou, Brandyn Castro, Catalina Lee-Chang, Maciej S. Lesniak

**Affiliations:** 1Department of Neurological Surgery, Feinberg School of Medicine, Northwestern University, Chicago, IL 60611, USA; mark.dapash@uchospitals.edu (M.D.); david.hou@northwestern.edu (D.H.); brandyn.castro@uchospitals.edu (B.C.); 2Pritzker School of Medicine, University of Chicago, Chicago, IL 60637, USA; 3Department of Neurosurgery, University of Chicago, Chicago, IL 60637, USA; 4Northwestern Medicine Malnati Brain Tumor Institute of the Lurie Comprehensive Cancer Center, Feinberg School of Medicine, Northwestern University, Chicago, IL 60611, USA

**Keywords:** glioblastoma, microenvironment, immunosuppression, blood-brain-barrier, tumor-associated myeloid cells, regulatory T cells, immune cells

## Abstract

GBM is the most common primary brain tumor in adults, and the aggressive nature of this tumor contributes to its extremely poor prognosis. Over the years, the heterogeneous and adaptive nature of GBM has been highlighted as a major contributor to the poor efficacy of many treatments including various immunotherapies. The major challenge lies in understanding and manipulating the complex interplay among the different components within the tumor microenvironment (TME). This interplay varies not only by the type of cells interacting but also by their spatial distribution with the TME. This review highlights the various immune and non-immune components of the tumor microenvironment and their consequences f the efficacy of immunotherapies. Understanding the independent and interdependent aspects of the various sub-populations encapsulated by the immune and non-immune components will allow for more targeted therapies. Meanwhile, understanding how the TME creates and responds to different environmental pressures such as hypoxia may allow for other multimodal approaches in the treatment of GBM. Ultimately, a better understanding of the GBM TME will aid in the development and advancement of more effective treatments and in improving patient outcomes.

## 1. Introduction

Occurring with an incidence of 3.19 per 100,000 persons, glioblastoma (GBM) represents the most common primary brain tumor in adults [1]. Even with the scientific advances that have allowed for greater knowledge of genomics, molecular biology, and more targeted therapies, patient outcomes remain poor, with a median survival following diagnosis of approximately 14 months [2]. The Stupp Protocol, comprising surgical resection plus adjuvant radiation therapy and temozolomide chemotherapy, followed by temozolomide has remained the standard of care since 2005 [3]. GBM patients with methylation of the O-6-methylguanine-DNA methyltransferase (MGMT) promoter, a DNA repair gene, have shown improved response to temozolomide chemotherapy, leading to improved patient outcomes [4]. The heterogeneity and adaptability of these tumors are the main contributors to their resistance to various therapeutic modalities. In 2010, data from The Cancer Genome Atlas (TCGA) were used to create a novel molecular classification system in which GBM could be group into proneural, classical, or mesenchymal subtypes. In 2010, four molecular subtypes of GBM were described as classical, mesenchymal, neural, and proneural [5]. Key genetic modifications were used to characterize these subtypes including epidermal growth factor receptor (EGFR) mutation/amplification/overexpression in the classical subtype, neurofibromin 1 (NF1) mutations/deletions in the mesenchymal subtype, EGFR amplification/overexpression in the neural subtype, and platelet-derived growth factor receptor alpha (PDGFRa) amplification in the proneural subtype [6]. A majority of GBM arise de novo from a specific trigger mutation in a glioma stem cell (GSC), known as primary GBMs. Secondary GBMs comprise a smaller percentage of GBMs and arise from lower-grade gliomas that acquire additional individual mutations. GBMs that arise from mutations in isocitrate dehydrogenase 1 (IDH1), commonly secondary GBMs, have a better prognosis than GBMs with wild-type IDH1 [7].

In addition to the previously mentioned genetic factors and mutational burden that affects prognosis, the GBM microenvironment contributes significantly to the dynamic and heterogeneous nature of GBM. The tumor microenvironment (TME) describes the active milieu of a tumor and is composed of stromal cells, signaling molecules, immune cells, and the surrounding extracellular matrix (ECM). The heterogeneous nature of GBM and the complex interplay among the different cell populations within its TME have wide-reaching implications (Figure 1). This dynamic interplay contributes to the establishment of hypoxic and necrotic tumor regions, infiltration into the surrounding parenchyma, resistance to radio-chemotherapy, and vascular proliferation [5]. Moreover, despite tumor infiltration via lymphocytes, GBM is considered to be a “cold tumor” due to its high amounts of regulatory B and T cells as well as immunosuppressive myeloid cells. Altogether, this creates a challenging setting for any immunotherapy to be fully efficacious against GBM, and there remains a need for a better understanding of the TME and its contributions towards the clinical picture seen in GBM. This review will summarize interactions within the TME and discuss their implications for the efficacy of immunotherapy.

## 2. The Extracellular Matrix

In a healthy individual, the extracellular matrix (ECM) is composed of various types of proteins and polysaccharides, which make up roughly 20% of the total volume in an adult human brain [8]. These macromolecules interact with neurons, astrocytes, and other cells to ultimately affect nearly all aspects of development and function. In the brain, various cell types contribute to ECM production, maturation, and structure, whereas ECM proteins in many other tissues are exclusively synthesized and deposited by fibroblasts and other mesenchymal cells [9]. In the setting of malignancies, the changes imposed by the tumor result in the ECM undergoing compositional changes based on their parent cell types. Typically, in the healthy human brain, the ECM is made up of a large proportion of proteoglycans (such as lecticans), glycoproteins (such as tenascin), and glycosaminoglycans (such as hyaluronan) as well as collagen and other fibrous matrix proteins [10]. In the setting of GBM, the composition of the ECM changes, which is physically reflected by the increased stiffness of the tumor-associated ECM [11]. Previous literature has highlighted that increased secretion of ECM components such as hyaluronic acid (HA), fibronectin, thrombospondin, and tenascin-C by glioma cells contribute to this change in ECM composition [12]. The increase of fibronectin and HA in the ECM, as well as the increased expression of particular receptors and integrins on the tumor cell, allow for increased mobility and invasiveness of glioma cells. For example, glioma cells can increase their expression of CD44, the main surface receptor for HA that also binds to matrix metalloproteinase 9 (MMP9) found in the ECM [13]. One specific component of the ECM, mesenchymal stromal cells (MSCs), play an important role in tumor migration. MSCs within the TME are able to release cytokines such as IL-6, CXCL1, and CXCL2, as well as metalloproteinases (MMPs), and contribute to the degradation of the local extracellular matrix [14,15]. Physical changes within the TME such as edema and cellular compression act as physical stressors for cells that cause increased stiffness of the tumor and can contribute to gliomagenesis. Interestingly, murine models have also demonstrated immunomodulation that occurs via the interaction of the innate immune system with the ECM of the tumor. The tumor ECM was shown to be intricately linked to CD47-mediated macrophage phagocytosis signaling through the expression of tumor-associated extracellular matrix protein tenascin C (TNC) [16].

## 3. The Blood–Brain Barrier

Another important non-immune component of the TME is the blood–brain barrier (BBB). The BBB is a unique attribute of the brain that allows for the tight regulation of molecules and cells [17]. The BBB is formed via the interaction of astrocyte foot processes with endothelial cells and pericytes. The BBB is considered one of the contributors to the poor chemotherapeutic efficacy often seen when treating brain malignancies with intravenous agents [18,19,20]. In the setting of GBM, the BBB is compromised not only because of inflammation and physical distortion but also because of the increased vascularity that contributes to the leakiness of blood vessels [21,22]. The dystrophic growth of the vasculature contributes to the heterogeneity of permeable vessel walls as well as to increased perfusion of the tumors. This increased angiogenesis is primarily due to the high amounts of vascular endothelial growth factor (VEGF) in the TME. Bevacizumab, an antibody that inhibits VEGF, was a therapy that initially showed promise against GBM but ultimately did not improve overall survival. Much of the false hope was due to the anti-angiogenic agent decreasing the leakiness of the vessel walls, leading to less gadolinium contrast within the tumors and the appearance of a smaller tumor on imaging. VEGF, interestingly, also causes the reduction of immune cell extravasation via the reduction in intracellular adhesion molecule 1 (ICAM1) and vascular cell adhesion molecule 1 (VCAM1) adhesion [23,24]. Ultimately, the BBB becomes compromised and leaky, allowing for the influx of some immune cells. Poor blood flow within the central necrotic areas of the tumors decreases oxygen delivery, contributing to the establishment of hypoxic regions that attract macrophages and promote their immunosuppressive phenotype [25,26].

## 4. The Central Nervous System Resident Cells

In addition to neurons, the glial compartment in the healthy adult brain comprises astrocytes, oligodendrocytes, and microglia, which serve various functions such as maintenance of the BBB, myelination of axons, and immune surveillance, respectively [27]. In the presence of GBM, the glial compartment within the TME undergoes numerous changes. Although oligodendrocyte-like cells have been found in many pathologic GBM sections, the extent to which these cells contribute to gliomagenesis requires further elucidation. During the growth of the tumor, healthy astrocytes are displaced by GBM-associated astrocytes, further contributing to the weakening of the BBB. Astrocytes within the GBM TME have been shown to undergo reactive astrogliosis similar to phenotypic changes undergone by astrocytes following a traumatic brain injury [28]. This reactive astrogliosis has been noted to contribute to tumor cell infiltration through the activation of zinc finger E-box-binding homeobox 1 (ZEB1), an epithelial–mesenchymal transition (EMT) transcription factor [29,30]. These tumor-associated astrocytes have also been highlighted for their involvement in modulating the immune system within the tumor microenvironment [14]. In fact, through the utilization of programmed death ligand 1 (PD-L1), tumor-associated astrocytes enhance immune suppression within the TME [28]. Microglia also exhibit an intricate interplay with cells encapsulated within the TME. Microglia are recruited to the TME by pro-migratory signals such as granulocyte-macrophage colony-stimulating factor (GM-CSF), stromal derived factor-1 (SDF-1), and glial cell line-derived neurotrophic factor (GDNF) secreted by tumor cells [31]. Microglia are further affected within the TME through their interactions with glioma cells. It was recently demonstrated that glioma cells use extracellular vesicles to interact directly with microglia [32]. This interaction causes genotypic and phenotypic changes that decrease their anti-tumor activity. Similarly, neurons, which also make up the glial compartment, have been shown to promote glioma progression [33,34]. Along with tumor-associated macrophages (TAMs), microglia largely contribute to immunosuppression through the interaction and synergistic release of soluble factors such as granulocyte-macrophage colony-stimulating factor (GM-CSF), C-X3-C Motif Chemokine Ligand 1, (CX3CL1), and SDF1 [35].

## 5. GBM Cells and Glioma Stem Cells

The communication between GBM cells and the TME is crucial for the proliferation, migration, and immunosuppression of the TME. One key chemokine utilized by tumor cells is C–C motif chemokine ligand 2 (CCL2), which functions to enhance angiogenesis and attract macrophages and microglia to the TME, further contributing to tumor growth [36]. GBM cells also secrete CXCL8, which can function to alter the ECM through the activation of MMPs within the TME [37,38]. GBM cells are also able to interact with microglia and increase their invasiveness by activating TGFβ and EGFR signaling pathways [39]. More recently, extracellular vesicles (EVs) have been highlighted as an important mechanism by which GBM cells communicate bidirectionally with the TME. EVs are used by GBM cells to interact with endothelial cells to promote angiogenesis and with astrocytes to promote ECM degradation [40]. EVs are also used by GBM cells to inhibit apoptosis of astrocytes, further contributing to the aggressiveness of the tumor [41]. Furthermore, EVs incorporating PD-L1 on their surface are able to inhibit T cell activation, further promoting the immunosuppressive environment of the TME [42]. Glioma stem cells (GSCs), like other cancer stem cells, serve as a reservoir for self-renewal and differentiation within the tumor. The differentiation into many unique cell lineages contributes to the heterogeneity seen in GBM and, consequently, to the decreased sensitivity to chemoradiotherapy [43,44,45,46]. In fact, subpopulations of GSCs within the TME have been shown to contribute to the differing susceptibility of GBMs to immunotherapy [47]. GSCs interact with endothelial cells, which results in an enhancement of stemness markers. These cells can be identified via several cell surface markers such as CD133, CD15/SSEA, CD44, and A2B5, although the heterogeneity of these cells prevents one marker from identifying and allow therapeutical targeting of all GSCs [48]. GSCs also contribute to the infiltrative nature of GBM, as studies have shown an association between the number of GSCs in the tumor bulk and the degree of invasiveness [49,50,51]. Their invasive capabilities are in fact enhanced in the presence of TAMs through TGF-β signaling.

## 6. Immune Cells

### 6.1. Tumor-Associated Myeloid Cells

A large portion of the tumor mass consists of immune cells. Tumor-associated myeloid cells (TAMCs) can make up as much as 50% of the tumor bulk [52]. TAMCs represent a heterogenous population composed of dendritic cells (DCs), neutrophils, bone marrow-derived macrophages (BMDMs), microglia, and myeloid-derived suppressor cells (MDSCs). It is thought that the majority of these tumor-infiltrating immune cells originate in the periphery rather than from the innate immune cells of the CNS [53]. Within the TAMC cellular compartment, TAMs are one of the most numerous subtypes, consisting of both microglia and bone-marrow-derived macrophages (BMDMs). The quantity of TAMs present within the TME correlates with tumor grade and inversely correlates with overall survival in patients with recurrent GBM [54,55]. TAMs can either release immunosuppressive factors such as interleukin 10 (IL-10) and transforming growth factor beta (TGF-β) or release anti-tumor-promoting factors such as IL-12, TNF-α, depending on the conditions within the TME [56,57]. Using a murine model, recent work has elucidated the phenotypic differences of TAMs based on their origins. TAMs derived from microglia are large, immobile cells with wide arrays of processes extending into the tumor, whereas TAMs derived from monocytes are small and mobile. These unique populations have been demonstrated in human GBM as well [35,58]. MDSCs are another heterogeneous population of cells that impart potent immunosuppressive effects on the TME. MDSCs can be divided into two subtypes: monocytic MDSCs (m-MDSCs) and polymorphonuclear MDSCs (PMN-MDSCs). Although similar in their immunosuppressive functions, each subtype has its own unique genomic profile [59]. M-MDSCs contribute to the overall pool of TAMs within the TME and aid in their aggregation within the tumor bulk. M-MDSCs within the hypoxic regions of the tumor undergo phenotypic changes that result in their differentiation into TAMs through the HIF-1α signaling pathway [60]. Similar to TAMs, increased quantities of MDSCs are correlated with the grade of the glioma as well as with a poor prognosis [61,62]. Likewise, at the time of recurrence, high percentages of MDSCs can be used as a poor predictive marker [63]. It has been highlighted that the majority of murine GBM-associated MDSCs in a murine model are M-MDSCs, yet the majority of MDSCs found in patient-derived GBMs are PMN-MDSCs [59]. A keyway in which MDSCs can suppress CD8^+^ T cell activity is through the increased catabolism of L-arginine. By using arginase-1 and inducible NO synthase, MDSCs utilize L-arginine in the surrounding TME, preventing T cells from utilizing this important amino acid and thus preventing proliferation [64]. They are also able to render the T cell receptor (TCR) non-functional by creating reactive oxygen species (ROS) that cause the nitration of the receptors [65]. Finally, MDSCs also contribute to the pool of IL-10 and TGF-β within the TME, further leading to immunosuppression.

### 6.2. Tumor-Associated Neutrophils

Neutrophils are another population of TAMCs that is found to accumulate within the GBM TME. Although not as numerous as MDSCs or TAMs, tumor-associated neutrophils (TANs) have been negatively associated with the prognosis of patients with GBM, and the quantity of TANs can serve as a negative prognostic marker for resistance to bevacizumab in patients who did not receive steroids [66]. TANs are typically found in the center of the tumor bulk and are attracted to the TME via macrophage migration inhibitory factor (MIF), C–X–C motif chemokine ligand 8 (CXCL8), and interleukin 8 (IL-8) [67]. TANs aid in tumor progression through their secretion of elastase, which functions to promote tumor proliferation and angiogenesis [68]. To a minor degree, TANs also contribute to the immunosuppressive TME via the secretion of arginase-1, granulocyte colony stimulating factor (G-CSF), and S100 calcium binding protein A4 (S100A4) [66,69].

### 6.3. Foxp3^+^ Regulatory T Cells

T cells play a vital role in the adaptive immune response to malignancies. Regulatory T cells (Tregs) are a unique population of T cells that serve to modulate the overall immune homeostasis through immunosuppressive measures. Of particular importance are Tregs that express Forkhead Box P3 (FOXP3) transcription factor. This transcription factor can downregulate the NFAT and NFκB signaling pathways, which consequently downregulates the expression of important effector cytokines such as IL2 [70,71]. A worse prognosis in GBM is associated with a higher Treg-to-T effector cell ratio [72,73]. Tregs are believed to be recruited to the TME via cytokines such as CXCL9/10/11-CXCR3 and CCL5-CCR5 [74]. These cytokines are secreted by innate immune cells within the CNS and glioma cells. Once within the TME, Tregs are subjected to favorable conditions which allow for increased viability and expansion, in addition to promoting the transition of other T cells into Tregs via cytokines such as tumor-derived IL-10 and TGF-β. Tregs themselves also secrete IL-10 and TGF-β [75,76,77] to further promote immunosuppression. These molecules are able to exert an immunosuppressive effect on natural killer (NK) cells, aid in the generation of MDSCs, and impair the antigen-presenting ability of DCs. Tregs also highly express key immune checkpoint molecules such as cytotoxic T lymphocyte-associated protein 4 (CTLA-4), programmed-death 1 (PD-1), and glucocorticoid-induced TNFR family-related gene (GITR) [78,79,80]. These molecules interact with their respective receptors on the surface of other immune cells to suppress the cells’ effector activities. Thus far, clinical trials investigating immune checkpoint inhibitors have mainly focused on targeting CTLA-4 and PD-1 in order modulate the anti-tumor response [81,82,83]. The use of T cells as immunotherapeutic tools has been explored in various malignancies including GBM. Chimeric antigen receptor (CAR) T cell therapy has been heavily investigated in several types of cancers [84,85,86,87]. These engineered T cells utilize tumor-associated antigens (TAAs) to allow T cells to become activated and gain greater specificity against tumor cells. TAAs such as variant III of the EGFR (EGFRvIII), human epidermal growth factor receptor 2 (HER2), or interleukin 13 (IL-13) receptor α2 (IL-13Rα2) have been investigated as possible targets [88,89,90,91]. More recent studies have investigated the efficacy of bispecific and trivalent CAR T cells [92]. Trivalent CAR T cells targeting HER2, IL13Rα2, and ephrin-A2 (EphA2) demonstrated improved cytotoxicity when compared to monospecific or bispecific CAR T cells [93]. Nevertheless, these T cells are subjected to the harsh TME of GBM and require further development.

### 6.4. Natural Killers

The innate immune system also contributes to the unique nature of the tumor microenvironment. NK cells are an important part of the innate immune system and are critical for the antitumor immune response, particularly through their interactions with major histocompatibility complex class I molecules (MHC-I). NK cells use granzyme B and perforin to provoke cellular apoptosis through contact-dependent cytotoxicity [94]. NK cells have been identified as part of the population of immune cells that infiltrate the GBM TME [95]. It has been shown that GBM with the R132H mutation in IDH1 contains neurons that can recruit NK cells to the CNS via the CX3CL1 chemokine [96]. NK cells have also been shown to be able to control tumor growth through cytokine secretion which is promoted via the NKp44 receptor. PDGF-D is expressed by most GBMs and binds to the activating NKp44 receptor to stimulate cytokine secretion from NK cells and innate lymphoid cells [97]. Tumor progression is also associated with B7-H6, which is known to augment NK cell functionality through the activation of their NKp30 receptors [98]. Nevertheless, NK cells are subjected to immunosuppressive factors within the TME. One key way by which the antitumor functionality of NK cells is suppressed is through cellular contact with glioma cells. Glioma cells can express unique MHC-I molecules that can bind to receptors on the surface of NK cells, thus suppressing their functions [99]. NK cells within the TME have been shown to regulate the levels of IFN-γ, which in turn can promote GSCs differentiation [100]. Furthermore, this change allows GSCs to become more resistant to NK cell cytotoxicity [101]. It has also been observed that radio-chemotherapy decreases the quantity of tumor-infiltrating NK cells [102]. The therapeutic potential of NK cells has been investigated by creating cytokine-induced killer (CIK) cells. CIK cells are created via culturing NK cells with IFN-γ, IL-2, and anti-CD3 monoclonal antibody (CD3 mAb) [103]. When tested in an open-labeled phase III clinical trial based in South Korea, this treatment did not show a significant difference in patient’s overall survival, highlighting the need for further investigation [104].

### 6.5. Dendritic Cells

Dendritic cells (DCs) are a class of antigen-presenting cells that serve as a vital bridge between the innate and the adaptive immune systems. DCs are important in monitoring pathogens or inflammatory responses throughout the body. DCs can endocytose, process, and present antigens to B and T cells, promoting their activation [105]. DCs are typically found in the meninges and the choroid plexus, but not within the healthy brain parenchyma [106]. In the setting of chronic inflammation, like that seen in GBM, DCs have been found within the brain. It has been shown in murine models that DCs are recruited to the TME via chemokines such as CCL5 and XCL1, similar to NK cells [107]. NK cells are also able to recruit DCs to the TME through the use of CCL5 and XCLI [107]. DCs have been shown to produce antitumor-augmenting cytokines such as IL12, which in turn can recruit more CD8^+^ T cells and reinvigorate anergic T cells [108]. Nevertheless, DCs are subjected to immunosuppressive effects from the TME which can induce a regulatory phenotype. These regulatory DCs can in turn activate Tregs and downregulate the recruitment of CD8^+^ T cells [109]. DCs have been a topic of interest for the development of new vaccines against GBM. Previous studies have highlighted the efficacy of DC-based vaccines in preclinical models as well as early-stage clinical trials [110,111,112]. Still, no successful phase III trials have been completed utilizing this type of vaccine.

### 6.6. B Cells

The role of B cells in antitumor immunity has remained nebulous in human and in in vivo studies. For one, tumor-infiltrating B cells have been correlated with poor prognosis of patients with metastatic carcinomas, while patients with breast carcinoma had improved survival [113,114]. Still, B cells are critical in their role as antigen-presenting cells and for their ability to induce clonal expansion of T cells sensitized to TAAs. B cells expressing 4-1BBL^+^ costimulatory surface protein have also been shown to be able to improve the functionality and viability of CD8^+^ T cells [115]. They do this by the secretion of cytokines such as TNFα [116]. Research efforts have now highlighted that regulatory B cells can also be found within the TME. Regulatory B cells were identified to overexpress immunosuppressive molecules such as PD-L1 and CD155 in addition to producing IL-10 and TGF-β [117] as a result of microvesicle-mediated interactions with MDSC. MDSCs were shown to secrete microvesicles containing PD-L1, which is then endocytosed by tumor-infiltrating B cells [52]. These B cells then act as potent inhibitors of CD8^+^ T cell cytotoxicity.

## 7. Implications of the TME on Immunotherapies

Immune checkpoint inhibitors (ICIs) are monoclonal antibodies which target inhibitory ligands and their receptors which are commonly expressed by GBM cells, lymphocytes, and myeloid cells [118]. Consequently, ICIs allow for a more robust antitumor response by CD8^+^ T cells. Most research into ICIs has focused on PD-1, PD-L1, and CTLA-4, important components of immune checkpoints pathways [78,80]. The use of ICIs has led to increased survival when compared to chemotherapy in patients with various tumors; however, it has had limited efficacy in the treatment of GBM in clinical trials [119,120,121]. Tumor vaccines utilize one or several tumor-associated antigens (TAAs) to stimulate a cellular and/or humoral immune response. Dendritic-based vaccines are created utilizing dendritic cells primed with whole cell tumor lysates or particular TAAs, while peptide-based vaccines activate T cells through the use of short amino acids [122]. For example, the variant III of the epidermal growth factor receptor (EGFRvIII) is a commonly acknowledged target for the use of peptide-based vaccination in GBM [123]. Clinical trials on both peptide- and dendritic-based vaccines have shown limited clinical benefits, despite the ability of these two strategies to invoke immune responses [112,124,125,126]. Another immunotherapeutic modality employs T cells with modified tumor-specific T cell receptors (TCRs) or chimeric antigen receptors (CARs) to generate a more precise anti-tumor response [127]. Currently, early-stage clinical trials involving CAR T cell therapy have shown safety and encouraging immune responses [128]. The various components of the GBM TME contribute to the limited efficacy of the current immunotherapeutic modalities. The non-immune components serve critical roles in the limited efficacy and eventual resistance through the creation of niches within the TME. The BBB and the irregular neovasculature prevent the optimal delivery of drugs, such as ICIs [129]. Furthermore, the inelastic and dense ECM increases metabolic stress and hypoxia, which ultimately activate pathways that inhibit apoptosis in cells closely interacting with the ECM [130]. GSCs are typically found within perivascular niches and closely interact with the ECM to modulate their function [131]. GCS contribute to tumor growth, invasion, and immunosuppression within the TME. The immunosuppressive nature of the TME remains one of the major limitations to achieving an effective anti-tumor response. Components such as regulatory B cells, regulatory T cells, and TAMCs secrete soluble factors such as IL-10 and TGF-β. This further promotes the recruitment and differentiation of additional regulatory B and T cells [132,133]. GBM cells are also able to interact with the TME, particularly the innate immune system, through the use of EVs containing PD-L1 on their surface [134]. These niches and complex interplay among the components of the TME create the intra- and inter-tumor heterogeneity which further limits the efficacy of immunotherapies.

## 8. Conclusions

The limited efficacy of the current treatments and the associated poor long-term prognosis of GBM is in a large part a consequence of the TME. This review highlights the intricate interplay among the immune and non-immune components of the TME. This interplay contributes to the establishment of a heterogeneous and adaptive TME which ultimately serves to increase the degree of immunosuppression, invasiveness, proliferation in these tumors. The non-immune components, particularly the BBB, as well as neurons, microglia, and ECM are important contributors to the alterations that take place within the TME. Still, the immune component made up of macrophages, DCs, B cells, and T cells can account for the majority of the extensive tumor-promoting effects seen within and outside of the TME. Several therapeutic approaches particularly targeting the TME have led to improved outcomes; however, this has been limited to very select sub-groups of patients. Immunotherapies such peptide-based and cell-based vaccines as well as immune checkpoint inhibitors aim to bolster the adaptive immune system to promote more robust anti-tumor responses. Yet, low tumor immunogenicity and immunosuppressive stressors, as a consequence of the interplay of various components of the TME, ultimately lead to resistance to immunotherapies. More recent work has allowed for further classification and identification of unique subpopulations of cells within the TME, further highlighting its heterogeneous nature. Continued investigation into the TME will aid in our understanding of how these elements contribute to the therapeutic response and interact with one another. This will lead to the creation of a more multimodal, yet targeted approach to the treatment of GBM.

## Figures and Tables

**Figure 1 cells-10-02257-f001:**
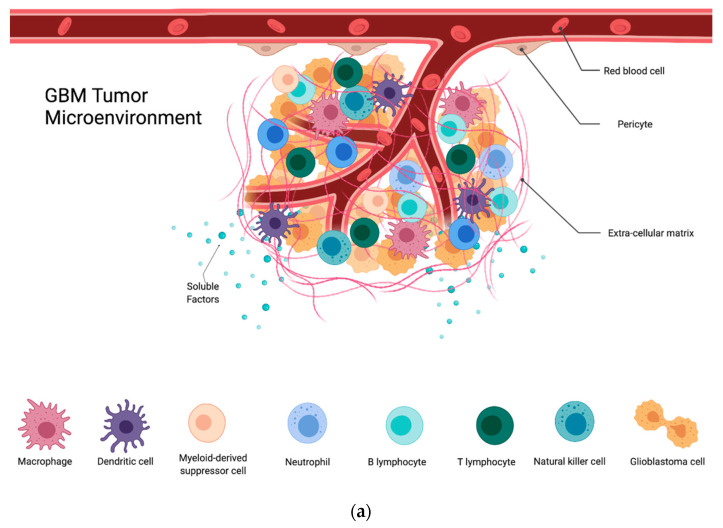
(**a**) GBM tumor microenvironment and its components; (**b**) components of the GBM TME and their contributions to the immunosuppression, proliferation, invasion, and treatment resistance seen in these tumors.

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
