# Peer review of "The Interplay between Glioblastoma and Its Microenvironment"

_cells, 2021, doi:10.3390/cells10092257_

Round 1

Reviewer 1 Report

In this review, Dapash and colleagues provide an integrative view of the role of tumor microenvironment in the progression of glioblastoma multiforme (GBM).

The authors highlight the functional interplay between the different cell components of the TME, both immune and non-immune, affecting the GBM characterization and prognosis. They also point out the limited efficacy of immunotherapies in this disease.

In my opinion, there should be a discussion section where the authors could better integrate and comment the contents of the previous sub-sections in order to provide a critical point of view on this issue with some perspective for future studies.

Also, poor emphasis is given to the secretome of glioblastoma cells into the extracellular milieu in the context of cell-to-cell communication during cell proliferation and invasion.

Finally, the authors should provide a representative figure depicting the interplay between the different cell types within the TME of GBM.

In general, the review is well conceptualized, it presents a concise but still exhaustive literature summary and is easy to read and understand.

Author Response

In this review, Dapash and colleagues provide an integrative view of the role of tumor microenvironment in the progression of glioblastoma multiforme (GBM).

The authors highlight the functional interplay between the different cell components of the TME, both immune and non-immune, affecting the GBM characterization and prognosis. They also point out the limited efficacy of immunotherapies in this disease.

In my opinion, there should be a discussion section where the authors could better integrate and comment the contents of the previous sub-sections in order to provide a critical point of view on this issue with some perspective for future studies.

The immunotherapy and limitations section was expanded to further discuss the current limitations of immunotherapies as a consequence of the individual components of the tumor microenvironment.

Also, poor emphasis is given to the secretome of glioblastoma cells into the extracellular milieu in the context of cell-to-cell communication during cell proliferation and invasion.

A separate section detailing GBM cells and glioma stem cells was created that further elaborated on how the secretome contributes to the proliferation and invasion of GBM tumors.

Finally, the authors should provide a representative figure depicting the interplay between the different cell types within the TME of GBM.

A two-part figure was created depicting the TME as well as the individual component and their respective contribution to the aggressive nature of TME

In general, the review is well conceptualized, it presents a concise but still exhaustive literature summary and is easy to read and understand.

Reviewer 2 Report

In this review article, Dapash and colleagues aim to review the various non-immune and immune components of the tumor microenvironment and their consequences on the efficacy of immunotherapies. The review article is broadly conceived and overall written well on basic scientific research.  However, content about therapeutic approaches targeting those components only described at lines 245 and 262 is weak. Other more readily addressable issues are listed below.

  1. The full name of indicated gene (e.g. IL-10, TGF-β, CXCL8) has not been added before the abbreviation. The abbreviation of TBI (traumatic brain injury) is not required.
  2. Descriptions of some components (e.g. the central nervous system resident cells, glioma stem cells, tumor-associated neutrophils) did not correlate to immunosuppression or immunotherapies.
  3. I believe that this review aims to collect literature on the efficacy of immunotherapies for GBM. More immunotherapeutic approaches particularly targeting the tumor microenvironment need to be included.

Author Response

In this review article, Dapash and colleagues aim to review the various non-immune and immune components of the tumor microenvironment and their consequences on the efficacy of immunotherapies. The review article is broadly conceived and overall written well on basic scientific research.  However, content about therapeutic approaches targeting those components only described at lines 245 and 262 is weak. Other more readily addressable issues are listed below.

1. The full name of indicated gene (e.g. IL-10, TGF-β, CXCL8) has not been added before the abbreviation. The abbreviation of TBI (traumatic brain injury) is not required.

The abbreviation for traumatic brain injury was removed and the appropriate names of indicated genes were included.

2. Descriptions of some components (e.g. the central nervous system resident cells, glioma stem cells, tumor-associated neutrophils) did not correlate to immunosuppression or immunotherapies.

The role of each of these components in the immunosuppressive nature of the TME was further expounded on.

3. I believe that this review aims to collect literature on the efficacy of immunotherapies for GBM. More immunotherapeutic approaches particularly targeting the tumor microenvironment need to be included.

The discussion section was expanded to further discuss the main types of immunotherapies currently investigated in the treatment of GBM. Furthermore, a new section detailing the limitations of these modalities due to individual components of the TME were included

Round 2

Reviewer 2 Report

The authors have provided a substantially improved manuscript to address some of the concerns from the prior submission. I have only a few minor comments left to address.

  1. The authors provide a new figure depicting the tumor microenvironment (TME) as well as the individual component and their respective contribution to the aggressive nature of TME but do not include it in the text. All figures must be mentioned in the text by their number.
  2. Typo: line 348, GCCs -> GSCs

Author Response

We would like to thank you for your time, dedication and constructive criticism which we believe contributed to a more comprehensive revised manuscript. We have addressed all the raised errors and hope this updated version of the manuscript fulfills the reviewers’ expectations.

Please find in italic our answer to your comment/question/concern.

  1. The authors provide a new figure depicting the tumor microenvironment (TME) as well as the individual component and their respective contribution to the aggressive nature of TME but do not include it in the text. All figures must be mentioned in the text by their number.

A reference to the figure was added to the introduction.

  1. Typo: line 348, GCCs -> GSCs

The typo was changed to correct abbreviation.